# RECURRENT NEURAL CIRCUITS
# FOR CONTOUR DETECTION

**Drew Linsley†, Junkyung Kim†‡, Alekh Ashok & Thomas Serre**
Department of Cognitive, Linguistic and Psychological Sciences
Brown University
Providence, RI 02912, USA
{drew_linsley,alekh_ashok,thomas_serre}@brown.edu
junkyung@google.com

## ABSTRACT

We introduce a deep recurrent neural network architecture that approximates known visual cortical circuits (Mély et al., 2018). We show that this architecture, which we refer to as the $\gamma$-Net, learns to solve contour detection tasks with better sample efficiency than state-of-the-art feedforward networks, while also exhibiting a classic perceptual illusion, known as the orientation-tilt illusion. Correcting this illusion significantly reduces $\gamma$-Net contour detection accuracy by driving it to prefer low-level edges over high-level object boundary contours. Overall, our study suggests that the orientation-tilt illusion is a byproduct of neural circuits that help biological visual systems achieve robust and efficient contour detection, and that incorporating such circuits in artificial neural networks can improve computer vision.

## 1 INTRODUCTION

An open debate since the inception of vision science concerns why we experience visual illusions. Consider the class of "contextual" illusions, where the perceived qualities of an image region, such as its orientation or color, are biased by the qualities of surrounding image regions. A well-studied contextual illusion is the orientation-tilt illusion depicted in Fig. 1a, where perception of the central grating's orientation is influenced by the orientation of the surrounding grating (O'Toole & Wenderoth, 1977). When the two orientations are similar, the central grating appears tilted slightly away from the surround (Fig. 1a, top). When the two orientations are dissimilar, the central grating appears tilted slightly towards the surround (Fig. 1a, bottom). Is the contextual bias of the orientation-tilt illusion a bug of biology or a byproduct of optimized neural computations?

Over the past 50 years, there has been a number of neural circuit mechanisms proposed to explain individual contextual illusions (reviewed in Mély et al., 2018). Recently, Mély et al. (2018) proposed a cortical circuit, constrained by physiology of primate visual cortex (V1), that offers a unified explanation for contextual illusions across visual domains – from the orientation-tilt illusion to color induction. These illusions arise in the circuit from recurrent interactions between neural populations with receptive fields that tile visual space, leading to contextual (center/surround) effects. For the orientation-tilt illusion, neural populations encoding the surrounding grating can either suppress or facilitate the activity of neural populations encoding the central grating, leading to repulsion vs. attraction, respectively. These surround neural populations compete to influence encodings of the central grating: suppression predominates when center/surround are similar, and facilitation predominates when center/surround are dissimilar.

The neural circuit of Mély et al. (2018) explains how contextual illusions might emerge, but it does not explain why. One possibility is that contextual illusions like the orientation-tilt illusion are "bugs": vestiges of evolution or biological constraints on the neural hardware. Another possibility is that contextual illusions are the by-product of efficient neural routines for scene segmentation (Keemink & van Rossum, 2016; Mély et al., 2018). Here, we provide computational evidence for the latter

---

† These authors contributed equally to this work.
‡ Currently at DeepMind, London, UK.

possibility and demonstrate that the orientation-tilt illusion reflects neural strategies optimized for object contour detection.

**Contributions**   We introduce the $\gamma$-Net, a trainable and hierarchical extension of the neural circuit of Mély et al. (2018), which explains contextual illusions. (i) The $\gamma$-Net is more sample efficient than state-of-the-art convolutional architectures on two separate contour detection tasks. (ii) Similar to humans but not state-of-the-art contour detection models, the $\gamma$-Net exhibits an orientation-tilt illusion after being optimized for contour detection. This illusion emerges from its preference for high-level object-boundary contours over low-level edges, indicating that neural circuits involved in contextual illusions also support sample-efficient solutions to contour detection tasks.

## 2   RELATED WORK

**Modeling the visual system**   Convolutional neural networks (CNNs) are often considered the *de facto* "standard model" of vision. CNNs and their extensions represent the state of the art for most computer vision applications with performance approaching – and sometimes exceeding – human observers on certain visual recognition tasks (He et al., 2016; Lee et al., 2017; Phillips et al., 2018). CNNs also provide the best fit to rapid neural responses in the visual cortex (see Kriegeskorte 2015; Yamins & DiCarlo 2016 for reviews). Nevertheless, multiple lines of evidence suggest that biological vision is still far more robust and versatile than CNNs (see Serre, 2019, for a recent review). CNNs suffer from occlusions and clutter (Fyall et al., 2017; Rosenfeld et al., 2018; Tang et al., 2018). They are also sample inefficient at learning visual relations (Kim et al., 2018) and solving simple grouping tasks (Linsley et al., 2018c). State-of-the-art CNNs require massive datasets to reach their impressive accuracy (Lake et al., 2015) and their ability to generalize beyond training data is limited (Geirhos et al., 2018; Recht et al., 2018).

Cortical feedback contributes to the robustness of biological vision (Hochstein & Ahissar, 2002; Wyatte et al., 2014; Kafaligonul et al., 2015). Feedforward projections in the visual system are almost always matched by feedback projections (Felleman & Van Essen, 1991), and feedback has been implicated in visual "routines" that cannot be implemented through purely feedforward vision, such as incremental grouping or filling-in (O'Reilly et al., 2013; Roelfsema, 2006). There is a also a growing body of work demonstrating the potential of recurrent neural networks (RNNs) to account for neural recordings (Fyall et al., 2017; Klink et al., 2017; Siegel et al., 2015; Tang et al., 2018; Nayebi et al., 2018; Kar et al., 2019; Kietzmann et al., 2019).

**Feedback for computer vision**   In contrast to CNNs, which build processing depth through a cascade of filtering and pooling stages with unique weights, RNNs process stimuli with filtering stages that reuse weights over "timesteps" of recurrence. On each discrete processing timestep, an RNN updates its hidden state through a nonlinear combination of an input and its the hidden state from its previous timestep. RNNs have been extended from their roots in sequence processing (e.g., Mozer 1992) to computer vision by computing the activity of RNN units through convolutional kernels. The common interpretation of these convolutional-RNNs, is that the input to each layer functions as a (fixed) feedforward drive, which is combined with layer-specific feedback from an evolving hidden state to dynamically adjust layer activity (Linsley et al., 2018c; George et al., 2017; Lotter et al., 2016; Wen et al., 2018; Liao & Poggio, 2016; Spoerer et al., 2017; Nayebi et al., 2018; Tang et al., 2018). In the current work, we are motivated by a similar convolutional-RNN, the horizontal gated recurrent unit (hGRU, Linsley et al. 2018a), which approximates the recurrent neural circuit model of (Mély et al., 2018) for explaining contextual illusions.

## 3   RECURRENT NEURAL MODELS

We begin by reviewing the dynamical neural circuit of Mély et al. (2018). This model explains contextual illusions by simulating interactions between cortical hypercolumns tiling the visual field (where hypercolumns describe a set of neurons encoding features for multiple visual domains at a single retinotopic position). In the model, hypercolumns are indexed by their 2D coordinate $(x, y)$ and feature channels $k$. Units in hypercolumns encode idealized responses for a visual domain (e.g., neural responses from the orientation domain were used to simulate an orientation-tilt illusion;

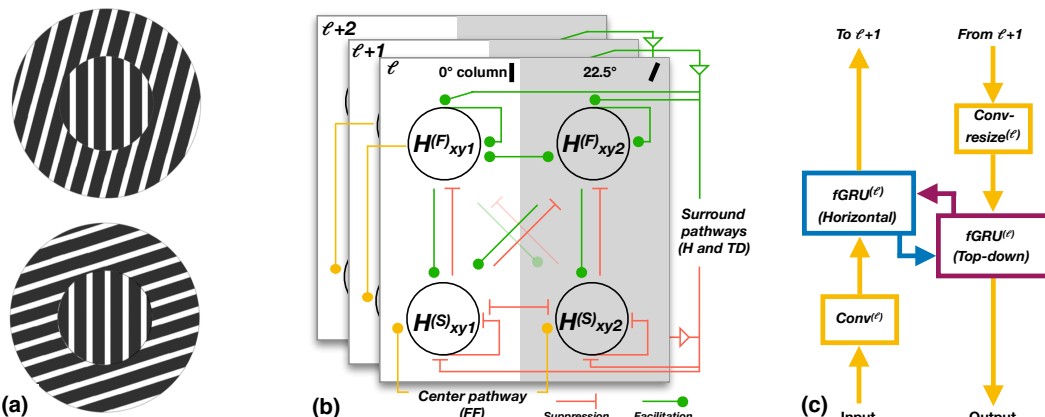

Figure 1: **The orientation tilt-illusion (O'Toole & Wenderoth, 1977) is a contextual illusion where a central grating's perceived orientation is influenced by a surround grating's orientation**. (a) When a central grating has a similar orientation as its surround, it is judged as tilting away from the surround (repulsion). When the two gratings have dissimilar orientations, the central grating is judged as tilting towards the surround (attraction). (b) We extend the recurrent circuit proposed by Mély et al. (2018) to explain this and other contextual illusions into a hierarchical model that learns horizontal (within a layer) and top-down (between layer) interactions between units. The circuit simulates dynamical suppressive ($\mathbf{H}^{(S)}_{xyk}$) and facilitative ($\mathbf{H}^{(F)}_{xyk}$) interactions between units in a layer $\ell$, which receives feedforward drive from a center pathway encoding feature k (e.g., edges oriented at 0° or 22.5°) at position (x, y) in an image. Blocks depict different layers, and arrowed connections denote top-down feedback. (c) A deep network schematic of the circuit diagram in (b), which forms the basis of the $\gamma$-Net introduced here. Horizontal and top-down connections are implemented with feedback gated recurrent units (fGRUs). Image encodings pass through these blocks on every timestep, from bottom-up (left path) to top-down (right path), and predictions are read out from the fGRU closest to image resolution on the final timestep. This motif can be stacked to create a hierarchical model.

Fig. 1b). Dynamics of a single unit at $xyk$ obey the following equations (we bold activity tensors to distinguish them from learned kernels and parameters):

$$\eta \dot{H}^{(S)}_{xyk} + \epsilon^2 H^{(S)}_{xyk} = \left[\xi Z_{xyk} - (\alpha H^{(F)}_{xyk} + \mu) C^{(S)}_{xyk}\right]_+ \qquad \text{\# Stage 1: Recurrent suppression of } \mathbf{Z}$$

$$\tau \dot{H}^{(F)}_{xyk} + \sigma^2 H^{(F)}_{xyk} = \left[\nu C^{(F)}_{xyk}\right]_+, \qquad \text{\# Stage 2: Recurrent facilitation of } \mathbf{H}^{(S)}$$

where

$$C^{(S)}_{xyk} = (W^S * \mathbf{H}^{(F)})_{xyk} \qquad \text{\# Compute suppression interactions}$$

$$C^{(F)}_{xyk} = (W^F * \mathbf{H}^{(S)})_{xyk}. \qquad \text{\# Compute facilitation interactions}$$

Circuit activities consist of a feedforward drive, recurrent suppression, and recurrent facilitation, respectively denoted as $\mathbf{Z}, \mathbf{H}^{(S)}, \mathbf{H}^{(F)} \in \mathbb{R}^{X \times Y \times K}$ ($X$ is width, $Y$ is height of the tensor, and $K$ is its feature channels)*. The circuit takes its "feedforward" input $\mathbf{Z}$ from hypercolumns (e.g., orientation encodings from hypercolumn units), and introduces recurrent suppressive and facilitatory interactions between units, $\mathbf{C}^{(S)}, \mathbf{C}^{(F)} \in \mathbb{R}^{X \times Y \times K}$ (Fig. 1b). These interactions are implemented with separate kernels for suppression and facilitation, $W^S, W^F \in \mathbb{R}^{E \times E \times K \times K}$, where $E$ is the spatial extent of connections on a single timestep (connectivity in this model is constrained by primate physiology).

---

*Suppression refers to interactions that reduce unit activity, and facilitation refers to interactions that increase activity. These computations can cause illusory repulsion or attraction at the level of neural populations.

These interactions are implemented through convolutions, allowing them to serially spread over timesteps of processing to connect units positioned at different spatial locations. The circuit outputs $\mathbf{H}^{(F)}$ after reaching steady state.

The circuit model of Mély et al. (2018) depends on competition between $\mathbf{H}^{(S)}$ and $\mathbf{H}^{(F)}$ to explain the orientation-tilt illusion. Competition is implemented by (i) computing suppression vs. facilitation in separate stages, and (ii) having non-negative activities, which enforces these functionally distinct processing stages. With these constraints in the circuit model, the strength of recurrent suppression – but not facilitation – multiplicatively increases with the net recurrent output. For the orientation-tilt illusion, suppression predominates when center and surround gratings have similar orientations. This causes encodings of the surround grating to "repulse" encodings of the center grating. On the other hand, facilitation predominates (causing "attraction") when center and surround gratings have dissimilar orientations because it is additive and not directly scaled by the circuit output.

Parameters controlling the circuit's integration, suppression/facilitation, and patterns of horizontal connections between units are tuned by hand. Linear and multiplicative suppression (i.e., shunting inhibition) are controlled by scalars $\mu$ and $\alpha$, feedforward drive is modulated by the scalar $\xi$, and linear facilitation is controlled by the scalar $\nu$. Circuit time constants are scalars denoted by $\eta, \epsilon, \tau$ and $\sigma$. All activities are non-negative and both stages are linearly rectified (ReLU) $[\cdot]_+ = \max(\cdot, 0)$.

**Feedback gated recurrent units**   Linsley et al. (2018a) developed a version of this circuit for computer vision applications, called the hGRU. In their formulation they use gradient descent (rather than hand-tuning like in the original circuit) to fit its connectivity and parameters to image datasets. The hGRU was designed to learn a difficult synthetic incremental grouping task, and a single layer of the hGRU learned long-range spatial dependencies that CNNs with orders-of-magnitude more weights could not. The hGRU replaced the circuit's time constants with dynamic gates, converted the recurrent state $\mathbf{H}^{(S)}$ for suppression into an instantaneous activity, and introduced a term for quadratic facilitation. The hGRU also relaxed biological constraints from the original circuit, including an assumption of non-negativity, which enforced competition between recurrent suppression vs. facilitation (e.g., guaranteeing that Stage 1 in the circuit model describes suppression of $Z_{xyk}$).

We extend the hGRU formulation in two important ways. First, like Mély et al. (2018), we introduce non-negativity. This constraint was critical for Mély et al. (2018) to explain contextual illusions, and as we describe below, was also important for our model. Second, we extend the circuit into a hierarchical model which can learn complex contour detection tasks. Recent neurophysiological work indicates that contextual effects emerge from both horizontal and top-down feedback (Chettih & Harvey, 2019). Motivated by this, we develop versions of the circuit to simulate horizontal connections between units within a layer, and top-down connections between units in different layers.

We call our module the feedback gated recurrent unit (fGRU). We describe the evolution of fGRU recurrent units in $\mathbf{H} \in \mathbb{R}^{X \times Y \times K}$, which are influenced by non-negative feedforward encodings $\mathbf{Z} \in \mathbb{R}^{X \times Y \times K}$ (e.g., a convolutional layer's response to a stimulus) over discrete timesteps $\cdot[t]$:

Stage 1:
$$\mathbf{G}^S = sigmoid(U^S * \mathbf{H}[t-1]) \qquad \text{\# Compute channel-wise selection}$$
$$\mathbf{C}^S = W^S * (\mathbf{H}[t-1] \odot \mathbf{G}^S) \qquad \text{\# Compute suppression interactions}$$
$$\mathbf{S} = \left[ \mathbf{Z} - \left[ (\alpha\mathbf{H}[t-1] + \mu)\,\mathbf{C}^S \right]_+ \right]_+ , \qquad \text{\# Suppression of } \mathbf{Z}$$

Stage 2:
$$\mathbf{G}^F = sigmoid(U^F * \mathbf{S}) \qquad \text{\# Compute channel-wise recurrent updates}$$
$$\mathbf{C}^F = W^F * \mathbf{S} \qquad \text{\# Compute facilitation interactions}$$
$$\tilde{\mathbf{H}} = \left[ \nu(\mathbf{C}^F + \mathbf{S}) + \omega(\mathbf{C}^F * \mathbf{S}) \right]_+ \qquad \text{\# Facilitation of } \mathbf{S}$$
$$\mathbf{H}[t] = (1 - \mathbf{G}^F) \odot \mathbf{H}[t-1] + \mathbf{G}^F \odot \tilde{\mathbf{H}}. \qquad \text{\# Update recurrent state}$$

Like the original circuit, the fGRU has separate stages for suppression (**S**) and facilitation (**H**). In the first stage, the feedforward encodings **Z** are suppressed by non-negative interactions between units in **H**[t − 1] (an fGRU hidden state from the previous timestep). Suppressive interactions are computed with the kernel $W^S \in \mathbb{R}^{E \times E \times K \times K}$, where $E$ describes the spatial extent of horizontal connections on a single timestep. This kernel is convolved with a gated version of the persistent hidden state **H**[t − 1]. The gate activity $\mathbf{G}^S$ is computed by applying a sigmoid nonlinearity to a convolution of the kernel $U^S \in \mathbb{R}^{1 \times 1 \times K \times K}$ with **H**[t − 1], which transforms its activity into the range [0, 1]. Additive and multiplicative forms of suppression are controlled by the parameters $\mu, \alpha \in \mathbb{R}^K$, respectively.

In the second stage, additive and multiplicative facilitation is applied to the instantaneous activity **S**. The kernels $W^F \in \mathbb{R}^{E \times E \times K \times K}$ controls facilitation interactions. Additive and multiplicative forms of facilitation are scaled by the parameters $\nu, \omega \in \mathbb{R}^K$, respectively. A gate activity is also computed during this stage to update the persistent recurrent activity **H**. The gate activity $\mathbf{G}^F$ is computed by applying a sigmoid to a convolution of the kernel $U^F \in \mathbb{R}^{1 \times 1 \times K \times \tilde{K}}$ with **S**. This gate updates **H**[t] by interpolating **H**[t − 1] with the candidate activity $\tilde{\mathbf{H}}$. After every timestep of processing, **H**[t] is taken as the fGRU output activity. As detailed in the following section, the fGRU output hidden state is either passed to the next convolutional layer (Fig. 1c, fGRU$^{(\ell)} \to$ conv$^{(\ell+1)}$), or used to compute top-down connections (Fig. 1c, fGRU$^{(\ell+1)} \to$ fGRU$^{(\ell)}$).

The fGRU has different configurations for learning horizontal connections between units within a layer or top-down connections between layers (Fig. 1b). These two configurations stem from changing the activities used for a fGRU's feedforward encodings and recurrent hidden state. "Horizontal connections" between units within a layer are learned by setting the feedforward encodings **Z** to the activity of a preceding convolutional layer, and setting the hidden state **H** to a persistent activity initialized as zeros (Fig. 1c, conv$^{(\ell)} \to$ fGRU$^{(\ell)}$). "Top-down connections" between layers are learned by setting fGRU feedforward encodings **Z** to the persistent hidden state $\mathbf{H}^{(\ell)}$ of a fGRU at layer $\ell$ in a hierarchical model, and the hidden state **H** to the persistent activity $\mathbf{H}^{(\ell+1)}$ of an fGRU at a layer one level up in the model hierarchy (Fig. 1c, fGRU$^{(\ell+1)} \to$ fGRU$^{(\ell)}$). The functional interpretation of the top-down fGRU is that it first suppresses activity in the lower layer using the higher layer's recurrent horizontal activities, and then applies a kernel to the residue for facilitation, which allows for computations like interpolation, sharpening, or "filling in". Note that an fGRU for top-down connections does not have a persistent state (it mixes high and low-level persistent states), but an fGRU for horizontal connections does.

$\gamma$**-Net**  Our main objective is to test how a model with the capacity for contextual illusions performs on natural image analysis. We do this by incorporating fGRUs into leading feedforward architectures for contour detection tasks, augmenting their feedforward processing with modules for learning feedback from horizontal and top-down connections (Fig. 1c). We refer to the resulting hierarchical models as $\gamma$-Nets, because information flows in a loop that resembles a $\gamma$: image encodings make a full bottom-up to top-down cycle through the architecture on every timestep, until dense predictions are read-out from the lowest-level recurrent layer of the network (thus, information flows in at the top of the hierarchy, loops through the network, and flows out from the top of the hierarchy). In our experiments we convert leading architectures for two contour detection problems into $\gamma$-Nets: A VGG16 for BSDS500 (He et al., 2019), and a U-Net for detection of cell membranes in serial electron microscopy images (Lee et al., 2017). See Appendix A for an algorithmic description of $\gamma$-net.

## 4 Contour detection experiments

**Overview**  We evaluated $\gamma$-Net performance on two contour detection tasks: object contour detection in natural images (BSDS500 dataset; Arbeláez et al., 2011) and cell membrane detection in serial electron microscopy (SEM) images of mouse cortex (Kasthuri et al., 2015) and mouse retina (Ding et al., 2016). Different $\gamma$-Net configurations were used on each task, with each building on the leading architecture for their respective datasets. All $\gamma$-Nets use 8-timesteps of recurrence and instance normalization (normalization controls vanishing gradients in RNN training; Ulyanov et al., 2016; Cooijmans et al., 2017, see Appendix A for details). The $\gamma$-Nets were trained with Tensorflow and NVIDIA Titan RTX GPUs using single-image batches and the Adam optimizer (Kingma & Ba, 2014, dataset-specific learning rates are detailed below). Models were trained with early stopping, which

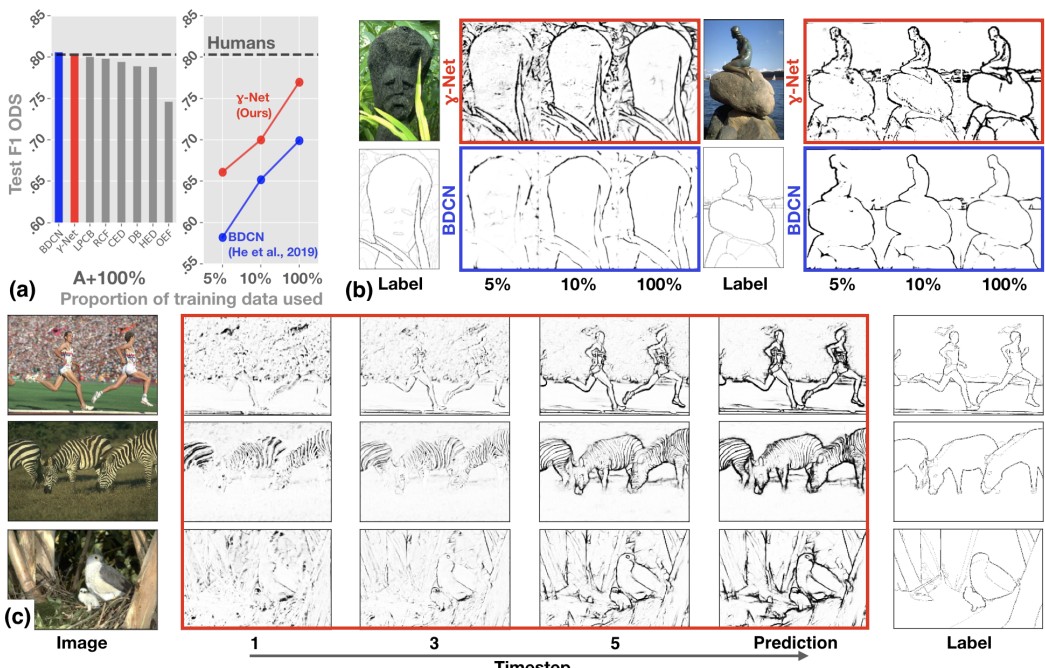

Figure 2: **Object contour detection in BSDS500 images. (a)** The $\gamma$-Net is on par with humans and the state-of-the-art for contour detection (BDCN; He et al. 2019) when trained on the entire training dataset with augmentations. In this regime, it also outperforms the published F1 ODS of all other approaches to BSDS500 (LPCB: Deng et al. 2018, RCF: Liu et al. 2019, CED: Wang et al. 2019, DB: Kokkinos 2015, HED: Xie & Tu 2017, and OEF: Hallman & Fowlkes 2015). The $\gamma$-Net outperforms the BDCN when trained on 5%, 10%, or 100% of the dataset. Performance is reported as F1 ODS (Arbeláez et al., 2011). **(b)** BDCN and $\gamma$-Net predictions after training on the different proportions of BSDS500 images. **(c)** The evolution of $\gamma$-Net predictions across timesteps of processing. Predictions from a $\gamma$-Net trained on 100% of BSDS are depicted: its initially coarse detections are refined over processing timesteps to select figural object contours.

terminated training if the validation loss did not drop for 50 straight epochs. The weights with the best validation-set performance were used for testing.

**Model evaluation**     We evaluated models in two ways. First, we validated them against state-of-the-art models for each contour dataset using standard benchmarks. As discussed below, we verified that our implementations of these state-of-the-art models matched published performance. Second, we tested sample-efficiency after training on subsets of the contour datasets without augmentations. Sample-efficiency compares the inductive biases of different architectures, and is critical for understanding how the capacity for exhibiting contextual illusions influences performance. We report model "wall time" in Appendix A; however, the focus of our work is on sample efficiency rather than the hardware/software-level optimizations that influence wall time. Model performance is evaluated as the F-measure at the Optimal Dataset Scale across images after non-maximum suppression post-processing (F1-ODS; Arbeláez et al., 2011), as is standard for contour detection tasks.

### 4.1 OBJECT CONTOUR DETECTION IN NATURAL IMAGES

**Dataset**     We trained models for object contour detection on the BSDS500 dataset (Arbeláez et al., 2011). The dataset contains object-contour annotations for 500 natural images, which are split into train (200), validation (100), and test (200) sets.

**Architecture details**     The leading approach to BSDS500 is the Bi-Directional Cascade Network (BDCN, He et al. 2019), which places multi-layer readout modules at every processing block in a

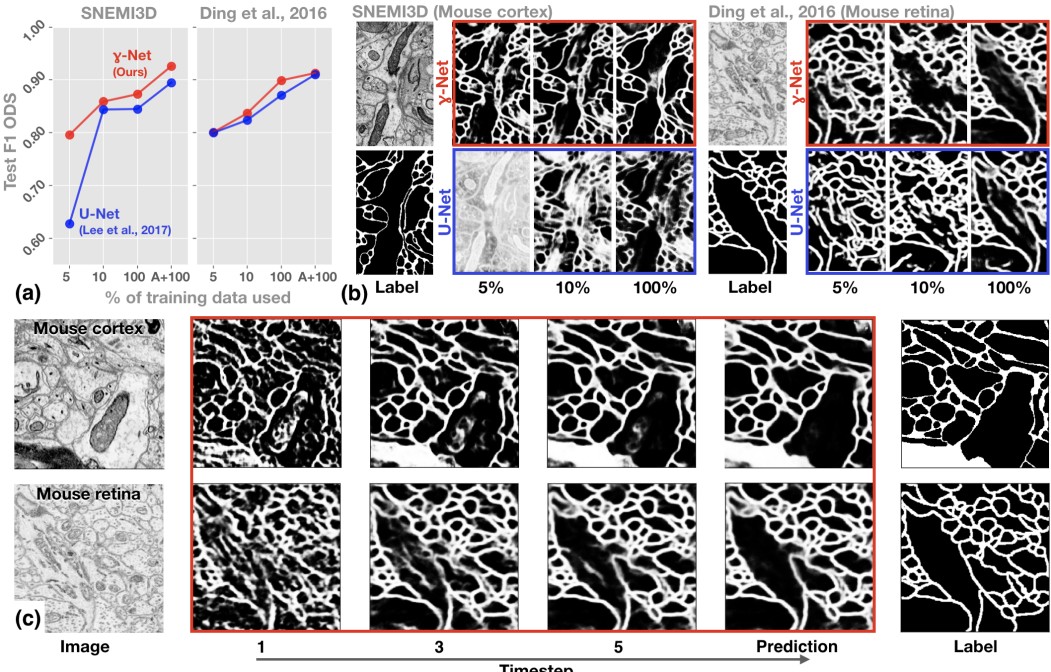

Figure 3: **Membrane prediction in serial electron microscopy (SEM) images of neural tissue**. **(a)** The $\gamma$-Net outperforms a state-of-the-art U-Net (Lee et al., 2017) for membrane detection when trained on SEM image datasets of mouse visual cortex (SNEMI3D) and retina (Ding et al., 2016). Performance is F1 ODS. **(b)** Network predictions after training on different proportions of each dataset. **(c)** The evolution of $\gamma$-Net predictions across timesteps of processing after training on 100% of the datasets. $\gamma$-Net learns to iteratively suppress contours belonging to internal cell features, such as organelles, which should not be annotated as contours for the purpose of neural tissue reconstruction.

ILSVRC12-pretrained VGG16, and optimizes a loss that balances contributions from each of these readouts to achieve better scale tolerance in final prediction. All leading deep learning approaches to BSDS500 begin with a VGG16 pretrained on ILSVRC12 object recognition (He et al., 2019).

Our $\gamma$-Net for BSDS500 begins with the same ILSVRC12-pretrained VGG16 as the BDCN. fGRUs were introduced for learning horizontal (conv2_2, conv3_3, conv4_3, conv5_3) and top-down connections (conv5_3 $\rightarrow$ conv4_3, conv4_3 $\rightarrow$ conv3_3, and conv3_3 $\rightarrow$ conv2_2). To pass top-down activities between layers, higher-level activities were resized to match lower-level ones, then passed through two layers of $1 \times 1$ convolutions with linear rectification, which registered feature representations from higher-to-lower layers. The $\gamma$-Net was trained with learning rates of $3e^{-4}$ on its randomly initialized fGRU weights and $1e^{-5}$ on its VGG-initialized weights. Training time and parameter counts for this $\gamma$-Net and the BDCN is in Appendix Table S1.

In contrast to the BDCN (and other recent approaches to BSDS) with multiple read-outs and engineered loss functions, we take $\gamma$-Net predictions as a linear transformation of the lowest fGRU layer in its feature hierarchy, and optimize the model with binary cross entropy between per-pixel predictions and labels (following the method of Xie & Tu, 2017). This approach works because the $\gamma$-Net uses feedback to merge high- and low-level image feature representations at the bottom of its feature hierarchy, resembling classic "V1 scratchpad" hypotheses for computation in visual cortex (Gilbert & Sigman, 2007; Lee & Mumford, 2003). We compared $\gamma$-Nets with a BDCN implementation released by the authors, which was trained using the routine described in He et al. (2019)[†].

---

[†]We replicated published results by following this routine and training the BDCN with the same regularization, batch size, and optimizer as He et al. (2019). For sample efficiency experiments, we searched through multiple learning rates and found that it had no affect on performance on these small BSDS500 datasets (Fig. S3).

**Results**  We validated the $\gamma$-Net against the BDCN after training on a full and augmented BSDS training set (Xie & Tu, 2017). The $\gamma$-Net performed similarly in F1 ODS (0.802) as the BDCN (0.806) and humans (0.803), and outperformed all other approaches to BSDS (Fig. 2a; Deng et al. 2018; Xie & Tu 2017; Hallman & Fowlkes 2015; Kokkinos 2015; Wang et al. 2019; Liu et al. 2019).

Our hypothesis is that contextual illusions reflect routines for efficient scene analysis, and that the capacity for exhibiting such illusions improves model sample efficiency. Consistent with this hypothesis, the $\gamma$-Net was up to an order-of-magnitude more efficient than the BDCN. A $\gamma$-Net trained on 5% of BSDS performs on par with a BDCN trained on 10% of the BSDS, and a $\gamma$-Net trained on 10% of the BSDS performs on par with a BDCN trained on 100% of BSDS. Unlike the BDCN, the $\gamma$-Net trained on 100% of BSDS outperformed the state of the art for non-deep learning based models (Hallman & Fowlkes, 2015), and nearly matched the performance of the popular HED trained with augmentations (Xie & Tu, 2017). We also evaluated lesioned versions of $\gamma$-Net to measure the importance of its horizontal/top-down connections, recurrence, non-negativity, and different specifications of its fGRU recurrent modules for detecting contours in BSDS500 (Fig. S4).

We examined recurrent feedback strategies learned by $\gamma$-Net for object contour by visualizing its performance on every timestep of processing. This was done by passing its activity at a timestep through the final linear readout layer. The $\gamma$-Net iteratively refines its initially coarse contour predictions. For example, the top row of Fig. 2c shows that the $\gamma$-Net selectively enhances the boundaries around the runner's bodies while suppressing the feature activities created by the crowd. In the next row of predictions, salient zebra stripes are gradually suppressed in favor of body contours (see Fig. S5 for $\gamma$-Net prediction dynamics and its tendency towards steady state solutions).

## 4.2 Cell membrane detection

**Datasets**  "Connectomics" involves extracting the wiring diagrams of neurons from serial electron microscope (SEM) imaging data, and is an important step towards understanding the algorithms of brains (Briggman & Bock, 2012). CNNs can automate this procedure by segmenting neuron membranes in SEM images. Large-scale challenges like SNEMI3D (Kasthuri et al., 2015), which contains annotated images of mouse cortex, have helped drive progress towards automation. Here, we test models on membrane detection in SNEMI3D and a separate SEM dataset of mouse retina ("Ding" from Ding et al. 2016). We split both datasets into training (80 images for SNEMI3D and 307 images for Ding) and test sets (20 images for SNEMI3D and 77 images for Ding). Next, we generated versions of each training dataset with 100%, 10%, or 5% of the images, as well as versions of the full datasets augmented with random left-right and up-down flips (A+100%).

**Architecture details**  The state-of-the-art on SNEMI3D is a U-Net variant (Ronneberger et al., 2015), which uses a different depth, different number of feature maps at every layer, and introduces new features like residual connections (Lee et al., 2017). We developed a $\gamma$-Net for cell membrane segmentation, which resembled this U-Net variant Lee et al. (2017) (see Appendix B for details). These $\gamma$-Net were trained from a random initialization with a learning rate of $1e^{-2}$ to minimize class-balanced per-pixel binary cross-entropy, and compared to the U-Net from Lee et al. (2017). Training time and parameter counts for these models is in Appendix Table S2.

**Results**  The $\gamma$-Net and U-Net of Lee et al. (2017) performed similarly when trained on full augmented versions of both the SNEMI3D and Ding datasets (Fig. 3a A+100%). However, $\gamma$-Nets were consistently more sample efficient than U-Nets on every reduced dataset condition (Fig. 3b).

We visualized the recurrent membrane detection strategies of $\gamma$-Nets trained on 100% of both datasets. Membrane predictions were obtained by passing neural activity at every timestep through the final linear readout. The $\gamma$-Net prediction timecourse indicates that it learns a complex visual strategy for membrane detection: it gathers a coarse "gist" of membranes in the first timestep of processing, and iteratively refines these predictions by enhancing cell boundaries and clearing out spurious contours of elements like cell organelles (Fig. 3c).

## 5 Bugs or byproducts of optimized neural computations?

**Orientation-tilt illusion**  Like the neural circuit of Mély et al. (2018), fGRU modules are designed with an asymmetry in their ability to suppress and facilitate feedforward input (see Section 3). This

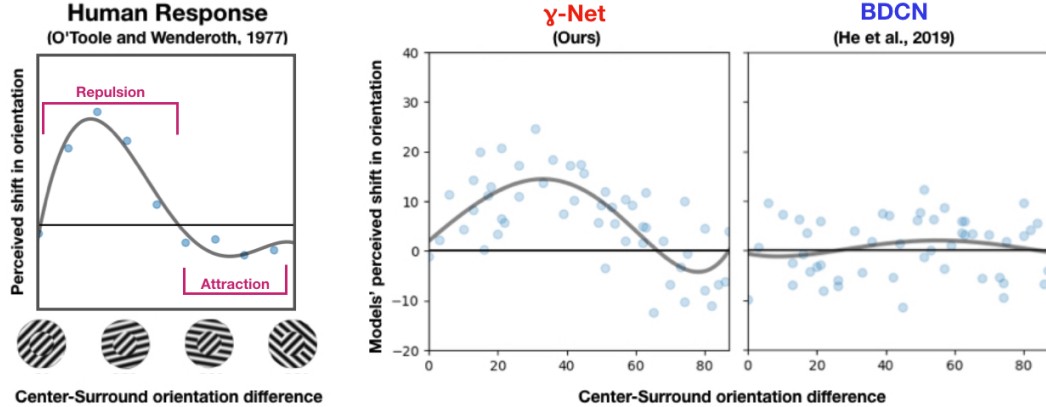

Figure 4: **Optimizing for contour detection produces an orientation-tilt illusion in the $\gamma$-Net.** The orientation-tilt illusion (O'Toole & Wenderoth, 1977) describes how perception of the center grating's orientation is repulsed from the surround when the two are in similar orientations (e.g., $\approx$ 30°), and attracted to the surround when the two are in dissimilar (but not orthogonal) orientations (e.g., $\approx$ 60°). We test for the orientation-tilt illusion in models trained on BSDS500 contour detection. Model weights were fixed and new layers were trained to decode the orientation of grating stimuli of a single orientation. These models were tested on grating stimuli in which surround orientations were systematically varied w.r.t. the center (exemplars depicted in the left panel). The $\gamma$-Net but not the BDCN had an orientation-tilt illusion. Gray curves depict a fourth-order polynomial fit.

potentially gives $\gamma$-Nets (which contain fGRU modules) the capacity to exhibit similar contextual illusions as humans. Here, we tested whether a $\gamma$-Net trained on contour detection in natural images exhibits an orientation-tilt illusion.

We tested for this illusion by training orientation decoders on the outputs of models trained on the full BSDS500 dataset. These decoders were trained on 100K grating images, in which the center and surround orientations were the same (Fig. S2a). These images were sampled from all orientations and spatial frequencies. The decoders had two 1×1 convolution layers and an intervening linear rectification to map model outputs into the sine and cosine of grating orientation. Both the $\gamma$-Net and BDCN achieved nearly perfect performance on a held-out validation set of gratings.

We tested these models on 1K grating stimuli generated with different center-surround grating orientations (following the method of O'Toole & Wenderoth 1977, Fig. S2b), and recorded model predictions for the center pixel in these images (detailed in Appendix C). Surprisingly, $\gamma$-Net encodings of these test images exhibited a similar tilt illusion as found in human perceptual data (Fig. 4b). There was repulsion when the central and surround gratings had similar orientations, and attraction when these gratings were dissimilar. This illusory phenomenon cannot be explained by accidental factors such as aliasing between the center and the surround, which would predict the opposite pattern, suggesting that the illusion emerges from the model's strategy for contour detection. In contrast, the BDCN, which only relies on feedforward processing, did not exhibit the effect (Fig. 4b). We also tested lesioned $\gamma$-Nets for the orientation-tilt illusion, but only the full $\gamma$-Net and a version lesioned to have only (spatially broad) horizontal connections replicated the entire illusion (thought this lesioned model performed worse on contour detection than the full model; Fig. S4). These findings are consistent with the work of Mély et al. (2018), who used spatially broad horizontal connections to model contextual illusions, as well as the more recent neurophysiological work of Chettih & Harvey (2019), who explained contextual effects through both horizontal and top-down interactions.

**Correcting the orientation-tilt illusion** What visual strategies does the orientation-tilt illusion reflect? We tested this question by taking a $\gamma$-Net that was trained to decode central grating orientation in tilt-illusion, and then training it further to decode the central grating orientation of *tilt-illusion stimuli* (Fig. 5a, "illusion-corrected" in red). Importantly, $\gamma$-Net weights were optimized during training, but the orientation decoder was not. Thus, improving performance for decoding the orientation of these illusory stimuli comes at the expense of changing $\gamma$-Net weights that were

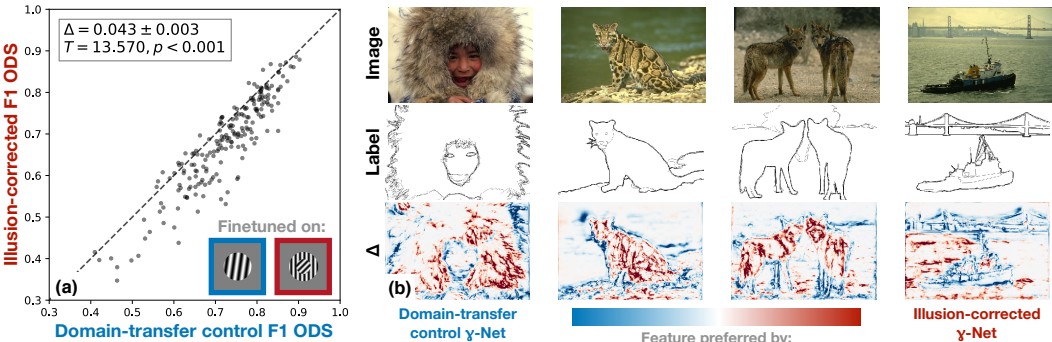

Figure 5: **Contour detection performance of the $\gamma$-Net depends on an orientation-tilt illusion.**
**(a)** F1 ODS scores on BSDS500 test images (200 total) from $\gamma$-Nets after correcting an orientation-tilt illusion ("illusion-corrected") or not ("domain-transfer control"). The domain-transfer control $\gamma$-Net was trained to decode the orientation of single-grating stimuli (blue), and the illusion-corrected $\gamma$-Net was trained to decode the orientation of the central grating in illusory grating stimuli (red). Readouts for decoding orientation were fixed, and $\gamma$-Net weights were allowed to change during training. Per-image F1 ODS was significantly greater for The domain-transfer control $\gamma$-Net than the illusion-correction $\gamma$-Net. **(b)** The illusion-corrected $\gamma$-Net was biased towards low-level contours, whereas the domain-transfer control $\gamma$-Net was biased towards contours on object boundaries.

responsible for its orientation-tilt illusion. As a control, another $\gamma$-Net was trained with the same routine to decode the orientation of full-image gratings, for which there is no illusion (Fig. 5a, "domain-transfer control" in blue; see Fig. S6 for training performance of both models). Both models were tested on the BSDS500 test set.

Correcting the orientation-tilt illusion of a $\gamma$-Net significantly hurts its object contour detection performance (Fig. 5a; 1-sample $T$-test of the per-image ODS F1 difference between models, $T(199) = 13.570$, $p < 0.001$). The illusion reflects $\gamma$-Net strategies for selecting object-boundaries rather than low-level contours (Fig. 5b; Fig. S7 for more examples).

# 6 CONCLUSION

Why do we experience visual illusions? Our experiments indicate that one representative contextual illusion, the orientation-tilt illusion, is a consequence of neural strategies for efficient scene segmentation. We directly tested whether this contextual illusion is a bug or a byproduct of optimized neural computations using the $\gamma$-net: a dense prediction model with recurrent dynamics inspired by neural circuits in visual cortex. On separate contour detection tasks, the $\gamma$-Net performed on par with state-of-the-art models when trained in typical regimes with full augmented datasets, but was far more efficient than these models when trained on sample-limited versions of the same datasets. At the same time, the $\gamma$-Net exhibited an orientation-tilt illusion which biased it towards high-level object-boundary contours over low-level edges, and its performance was reduced when it was trained to correct its illusion.

While $\gamma$-Nets are more sample efficient than leading feedforward models for contour detection, they also take much more "wall-time" to train than these feedforward models. Learning algorithms and GPU optimizations for RNNs are lagging behind their feedforward counterparts in computational efficiency, raising the need for more efficient approaches for training hierarchical RNNs like $\gamma$-Nets to unlock their full potential.

More generally, our work demonstrates novel synergy between artificial vision and vision neuroscience: we demonstrated that circuit-level insights from biology can improve the sample efficiency of deep learning models. The neural circuit that inspired the fGRU module explained biological illusions in color, motion, and depth processing (Mély et al., 2018), and we suspect that $\gamma$-Nets will have similar success in learning sample-efficient strategies – and exhibiting contextual illusions – in these domains.

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

# A  γ-NET

---

**Algorithm 1** A generic γ-Netarchitecture. See Section 3 in the main text for the treatment.

---

**Require:** Image batch $\mathbf{I}$

1: $\mathbf{Z}^\ell \leftarrow \mathbf{I}$
2: $\mathbf{H}^0[t] \leftarrow 0$
3: **for** $t = T$ **do**                                              ▷ Recurrent loops over static images
4:    **for** $\ell = 0$ to L **do**                                  ▷ Bottom-up pass
5:       $\mathbf{Z}^\ell = \text{ReLU}(\text{Conv}(\mathbf{Z}^\ell))$   ▷ Feedforward activity
6:       $\mathbf{H}^\ell[t] = fGRU_H(Z{=}\mathbf{Z}^\ell, H{=}\mathbf{H}^\ell[t-1])$   ▷ Update horizontal feedback state
7:       **if** $\ell < $ L **then**
8:          $\mathbf{Z}^{\ell+1} = \text{Maxpool}(\mathbf{H}^\ell[t])$
9:    **for** $\ell = L$ to 0 **do**                                  ▷ Top-down feedback pass
10:      **if** $\ell < L$ **then**
11:         $\mathbf{H}^\ell[t] = \text{fGRU}_{\text{TD}}(Z{=}\mathbf{H}^\ell[t], H{=}\text{ReLU}(\text{Conv}(\text{Upsample}(\mathbf{H}^{\ell+1}[t]))))$
12: **return** $\mathbf{H}^0[t]$                                      ▷ Hidden state of layer $\ell$

---

**Connectomics**   The current standard for computer vision applications in connectomics is to train and test on separate partitions of the same tissue volume (Linsley et al., 2018b; Januszewski & Jain, 2019). This makes it difficult to develop new model architectures without overfitting to any particular dataset. For this reason, we first tuned our connectomics γ-Net and its hyperparameters on a synthetic dataset of cell images (data not shown).

In our experiments on synthetic data, we noted monotonically improved performance with increasing timesteps, which motivated our choice of building these models with as many timesteps as could fit into GPU memory without sacrificing (we carried this concept over to the design of our γ-Nets for BSDS). Thus, we settled on 8 timesteps for the γ-Nets . We also compared our use of fGRU modules to learn recurrent connection vs. the classic LSTM and GRU recurrent modules, and found that the γ-Netwas far more effective on small datasets, which we take as evidence that its recurrent application of suppression separately from facilitation is a better inductive bias for learning contour tasks (see Hahnloser et al. 2000 for a theoretical discussion on how these operations can amount to a digital selection of task-relevant features through inhibition, followed by an analog amplification of the residuals through excitation).

We found that γ-Net cell membrane detection was improved when every bottom-up unit (from a typical convolution) was given a hidden state. Like with gated recurrent architectures, these gates enable gradients to effectively skip timesteps of processing where they pathologically decay. We do this by converting every convolutional layer (except the first and last) into a "minimal gated unit" (Heck & Salem 2017). This conversion introduced two additional kernels to each convolutional layer, $U^F, W^H \in \mathbb{R}^{1\times1\times K\times K}$, where the former was responsible for selecting channels from a persistent activity $\mathbf{H} \in \mathbb{R}^{X\times Y\times K}$ for processing on a given timestep and updating the persistent activity. The latter kernel transformed a modulated version of the hidden state $\mathbf{H}$. This transformed hidden state was combined with a vanilla convolutional feedforward encoding, $\mathbf{Z} \in \mathbb{R}^{X\times Y\times K}$ (see Eq. 1 for the treatment). Weights in these layers were initialized with orthogonal random initializations, which help training recurrent networks (Vorontsov et al., 2017).

$$\begin{aligned}
\mathbf{F} &= \sigma(\mathbf{Z} + W^F * \mathbf{H}[t-1] + \mathbf{b}_F) \\
\mathbf{H}[t] &= \mathbf{F} \odot \mathbf{H}[t-1] + (1 - \mathbf{F}) \odot \text{ELU}(Z + W^H * (\mathbf{F} \odot \mathbf{H}[t-1]) + \mathbf{b}_H)
\end{aligned} \tag{1}$$

**fGRU**   Here we describe additional details of the fGRU. fGRU kernels for computing suppressive and facilitative interactions have symmetric weights between channels, similar to the original circuit of Mély et al. (2018). This means that the weight $W_{x_0+\Delta x, y_0+\Delta y, k_1, k_2}$ is equal to the weight $W_{x_0+\Delta x, y_0+\Delta y, k_2, k_1}$, where $x_0$ and $y_0$ denote kernel center. This constraint means that there are nearly half as many learnable connections as a normal convolutional kernel. In our experiments, this constraint improved performance.

**BSDS** (20M parameters) $\gamma$-**Net**

| Layer | Operation | Output shape |
|---|---|---|
| conv-1-down | conv 3 × 3 / 1 | 320 × 480 × 64 |
| | conv 3 × 3 / 1 | 320 × 480 × 64 |
| | maxpool 2 × 2 / 2 | 160 × 240 × 64 |
| conv-2-down | conv 3 × 3 / 1 | 160 × 240 × 128 |
| | conv 3 × 3 / 1 | 160 × 240 × 128 |
| | fGRU-horizontal 3 × 3 / 1 | 160 × 240 × 128 |
| | maxpool 2 × 2 / 2 | 80 × 120 × 128 |
| conv-3-down | conv 3 × 3 / 1 | 80 × 120 × 256 |
| | conv 3 × 3 / 1 | 80 × 120 × 256 |
| | conv 3 × 3 / 1 | 80 × 120 × 256 |
| | fGRU-horizontal 3 × 3 / 1 | 80 × 120 × 256 |
| | maxpool 2 × 2 / 2 | 40 × 60 × 256 |
| conv-4-down | conv 3 × 3 / 1 | 40 × 60 × 512 |
| | conv 3 × 3 / 1 | 40 × 60 × 512 |
| | conv 3 × 3 / 1 | 40 × 60 × 512 |
| | fGRU-horizontal 3 × 3 / 1 | 40 × 60 × 512 |
| | maxpool 2 × 2 / 2 | 20 × 30 × 512 |
| conv-5-down | conv 3 × 3 / 1 | 20 × 30 × 512 |
| | conv 3 × 3 / 1 | 20 × 30 × 512 |
| | conv 3 × 3 / 1 | 20 × 30 × 512 |
| | fGRU-horizontal 3 × 3 / 1 | 20 × 30 × 512 |
| conv-4-up | instance-norm | 20 × 30 × 512 |
| | bilinear-resize | 40 × 60 × 512 |
| | conv 1 × 1 / 1 | 40 × 60 × 8 |
| | conv 1 × 1 / 1 | 40 × 60 × 512 |
| | fGRU-top-down 1 × 1 / 1 | 40 × 60 × 512 |
| conv-3-up | instance-norm | 40 × 60 × 512 |
| | bilinear-resize | 80 × 120 × 512 |
| | conv 1 × 1 / 1 | 80 × 120 × 16 |
| | conv 1 × 1 / 1 | 80 × 120 × 256 |
| | fGRU-top-down 1 × 1 / 1 | 80 × 120 × 256 |
| conv-2-up | instance-norm | 80 × 120 × 256 |
| | bilinear-resize | 160 × 240 × 256 |
| | conv 1 × 1 / 1 | 160 × 240 × 64 |
| | conv 1 × 1 / 1 | 160 × 240 × 128 |
| | fGRU-top-down 1 × 1 / 1 | 160 × 240 × 128 |
| Readout | instance-norm | 160 × 240 × 128 |
| | bilinear-resize | 320 × 480 × 128 |
| | conv 1 × 1 / 1 | 320 × 480 × 1 |

Table S1: $\gamma$-Net architecture for contour detection in BSDS natural images. For comparison, the BDCN, which is the state of the art on BSDS, contains ≈16.3M parameters. When training on an NVIDIA GeForce RTX, this $\gamma$-Nettakes 1.8 seconds per image, whereas the BDCN takes 0.1 seconds per image. "Down" refers to down-sampling layers; "up" refers to up-sampling layers, and "readout" maps model activities into per-pixel decisions. Kernels are described as kernel-height × kernel-width / stride size. All convolutional layers except for the Readout use non-linearities. All non-linearities in this network are linear rectifications. Model predictions come from the fGRU hidden state for conv-2-down, which are resized to match the input image resolution and passed to the linear per-pixel readout.

| Connectomics $\gamma$-Net (450K parameters) | | |
|---|---|---|
| **Layer** | **Operation** | **Output shape** |
| conv-1-down | conv $3 \times 3$ / 1 | $384 \times 384 \times 24$ |
| | conv $3 \times 3$ / 1 | $384 \times 384 \times 24$ |
| | fGRU-horizontal $9 \times 9$ / 1 | $384 \times 384 \times 24$ |
| | maxpool $2 \times 2$ / 2 | $192 \times 192 \times 24$ |
| conv-2-down | conv $3 \times 3$ / 1 | $192 \times 192 \times 28$ |
| | fGRU-horizontal $7 \times 7$ / 1 | $192 \times 192 \times 28$ |
| | maxpool $2 \times 2$ / 2 | $96 \times 96 \times 28$ |
| conv-3-down | conv $3 \times 3$ / 1 | $96 \times 96 \times 36$ |
| | fGRU-horizontal $5 \times 5$ / 1 | $96 \times 96 \times 36$ |
| | maxpool $2 \times 2$ / 2 | $48 \times 48 \times 36$ |
| conv-4-down | conv $3 \times 3$ / 1 | $48 \times 48 \times 48$ |
| | fGRU-horizontal $3 \times 3$ / 1 | $48 \times 48 \times 48$ |
| | maxpool $2 \times 2$ / 2 | $24 \times 24 \times 48$ |
| conv-5-down | conv $3 \times 3$ / 1 | $24 \times 24 \times 64$ |
| | fGRU-horizontal $1 \times 1$ / 1 | $24 \times 24 \times 64$ |
| conv-4-up | transpose-conv $4 \times 4$ / 2 | $48 \times 48 \times 48$ |
| | conv $3 \times 3$ / 1 | $48 \times 48 \times 48$ |
| | instance-norm | $48 \times 48 \times 48$ |
| | fGRU-top-down $1 \times 1$ / 1 | $48 \times 48 \times 48$ |
| conv-3-up | transpose-conv $4 \times 4$ / 2 | $96 \times 96 \times 36$ |
| | conv $3 \times 3$ / 1 | $96 \times 96 \times 36$ |
| | instance-norm | $96 \times 96 \times 36$ |
| | fGRU-top-down $1 \times 1$ / 1 | $96 \times 96 \times 36$ |
| conv-2-up | transpose-conv $4 \times 4$ / 2 | $192 \times 192 \times 28$ |
| | conv $3 \times 3$ / 1 | $192 \times 192 \times 28$ |
| | instance-norm | $192 \times 192 \times 28$ |
| | fGRU-top-down $1 \times 1$ / 1 | $192 \times 192 \times 28$ |
| conv-1-up | transpose-conv $4 \times 4$ / 2 | $384 \times 384 \times 24$ |
| | conv $3 \times 3$ / 1 | $384 \times 384 \times 24$ |
| | instance-norm | $384 \times 384 \times 24$ |
| | fGRU-top-down $1 \times 1$ / 1 | $384 \times 384 \times 24$ |
| Readout | instance-norm | $384 \times 384 \times 24$ |
| | conv $5 \times 5$ / 1 | $384 \times 384 \times 24$ |

Table S2: $\gamma$-Netarchitecture for cell membrane detection in SEM images. A 2D version of the U-Net of Lee et al. (2017), which is the state of the art on SNEMI3D, contains $\approx$600K parameters. When training on an NVIDIA GeForce RTX, this $\gamma$-Nettakes 0.7 seconds per image, whereas the U-Net takes 0.06 seconds per image. "Down" refers to down-sampling layers; "up" refers to up-sampling layers, and "readout" maps model activities into per-pixel decisions. Kernels are described as kernel-height $\times$ kernel-width / stride size. All fGRU non-linearities are linear rectifications, and all convolutional non-linearities are exponential linear units (ELU), as in (Lee et al., 2017). All convolutional layers except for the Readout use non-linearities. Model predictions come from the fGRU hidden state for conv-1-down, which are passed to the linear readout.

While optimizing $\gamma$-Nets on synthetic cell image datasets, we found that a small modification of the fGRU input gate offered a modest improvement in performance. We realized that the input gate in the fGRU is conceptually similar to recently developed models for feedforward self-attention in deep neural networks. Specifically, the global-and-local attention modules of (Linsley et al., 2019), in which a non-linear transformation of a layer's activity is used to modulate the original activity. Here, we took inspiration from global-and-local attention, and introduced an additional gate into the fGRU, resulting in the following modification of the main equations.

Stage 1:

$$\mathbf{A}^S = U^A * \mathbf{H}[t-1] \qquad \text{\# Compute channel-wise selection}$$

$$\mathbf{M}^S = U^M * \mathbf{H}[t-1] \qquad \text{\# Compute spatial selection}$$

$$\mathbf{G}^S = sigmoid(IN(\mathbf{A}^S \odot \mathbf{M}^{S*})) \qquad \text{\# Compute suppression gate}$$

$$\mathbf{C}^S = IN(W^S * (\mathbf{H}[t-1] \odot \mathbf{G}^S)) \qquad \text{\# Compute suppression interactions}$$

$$\mathbf{S} = \left[\mathbf{Z} - \left[(\alpha\mathbf{H}[t-1] + \mu)\,\mathbf{C}^S\right]_+\right]_+, \qquad \text{\# Additive and multiplicative suppression of } \mathbf{Z}$$

Stage 2:

$$\mathbf{G}^F = sigmoid(IN(U^F * \mathbf{S})) \qquad \text{\# Compute channel-wise recurrent updates}$$

$$\mathbf{C}^F = IN(W^F * \mathbf{S}) \qquad \text{\# Compute facilitation interactions}$$

$$\tilde{\mathbf{H}} = \left[\nu(\mathbf{C}^F + \mathbf{S}) + \omega(\mathbf{C}^F * \mathbf{S})\right]_+ \qquad \text{\# Additive and multiplicative facilitation of } \mathbf{S}$$

$$\mathbf{H}[t] = (1 - \mathbf{G}^F) \odot \mathbf{H}[t-1] + \mathbf{G}^F \odot \tilde{\mathbf{H}} \qquad \text{\# Update recurrent state}$$

$$\text{where } IN(\mathbf{r}; \delta, \nu) = \nu + \delta \odot \frac{\mathbf{r} - \widehat{\mathbb{E}}[\mathbf{r}]}{\sqrt{\widehat{\text{Var}}[\mathbf{r}] + \eta}}.$$

This yields the global input gate activity $\mathbf{A}^S \in \mathbb{R}^{X \times Y \times K}$ and the local input gate activity $\mathbf{M}^{S*} \in \mathbb{R}^{X \times Y \times 1}$, which are computed as filter responses between the previous hidden state $\mathbf{H}[t-1]$ and the global gate kernel $U^A \in \mathbb{R}^{1 \times 1 \times K \times K}$ and the local gate kernel $U^M \in \mathbb{R}^{3 \times 3 \times K \times 1}$. Note that the latter filter is learning a mapping into 1 dimension and is therefore first tiled into $K$ dimensions, yielding $\mathbf{M}^{S*}$, before elementwise multiplication with $\mathbf{A}^S$. All results in the main text use this implementation.

Following the lead of (Linsley et al., 2018a), we incorporated normalizations into the fGRU. Let $\mathbf{r} \in \mathbb{R}^d$ denote the vector of layer activations that will be normalized. We chose instance normalization (Ulyanov et al., 2016) since it is independent of batch size, which was 1 for $\gamma$-Nets in our experiments. Instance normalization introduces two $k$-dimensional learned parameters, $\delta, \nu \in \mathbb{R}^d$, which control the scale and bias of normalized activities, and are are shared across timesteps of processing. In contrast, means and variances are computed on every timestep, since fGRU activities are not i.i.d. across timesteps. Elementwise multiplication is denoted by $\odot$ and $\eta$ is a regularization hyperparameter.

Learnable gates, such as those in the fGRU, are helpful for training RNNs. But there are other heuristics that are also important for optimizing performance. We use several of these with $\gamma$-Nets, such as Chronos initialization of fGRU gate biases (Tallec & Ollivier, 2018) and random orthogonal initialization of kernels (Vorontsov et al., 2017). We initialized the learnable scale parameter $\delta$ of fGRU normalizations to 0.1, since values near 0 optimize the dynamic range of gradients passing through its sigmoidal gates (Cooijmans et al., 2017). Similarly, fGRU parameters for learning additive suppression/facilitation $(\mu, \nu)$ were initialized to 0, and parameters for learning multiplicative inhibition/excitation $(\alpha, \omega)$ were initialized to 0.1. Finally, when implementing top-down connections, we incorporated an extra skip connection. The activity of layer $\ell$ was added to the fGRU-computed top-down interactions between layer $\ell$ and layer $\ell + 1$. This additional skip connection improved the stability of training.

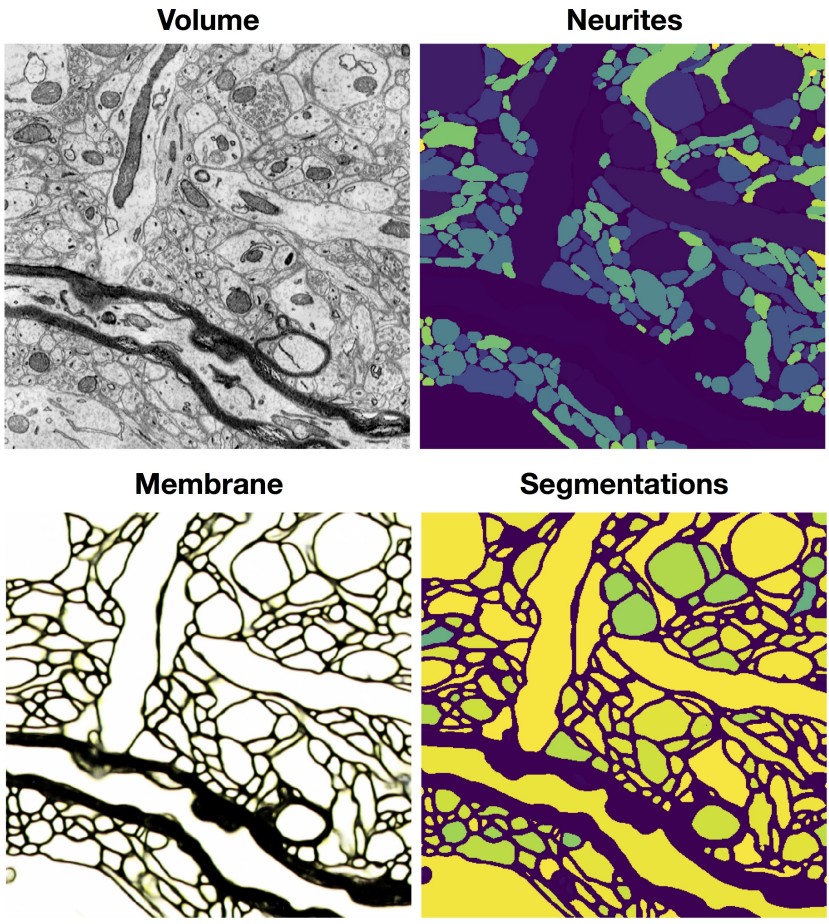

Figure S1: We trained the reference 3D U-Net from (Lee et al., 2017) on the SNEMI3D dataset to validate the implementation. Segmentations here are derived by watershedding and agglomeration with GALA (Nunez-Iglesias et al., 2013), resulting in "superhuman" ARAND (evaluated according to the SNEMI3D standard; lower is better) of 0.04, which is below the reported human-performance threshold of 0.06 and on par with the published result (see Table 1 in Lee et al. 2017, mean affinity agglomeration).

## B    MEMBRANE PREDICTION MODELS

Our reference model for membrane prediction is the 3D U-Net of (Lee et al., 2017). This architecture consists of four encoder blocks (multiple convolutions and skip connections, pooling and subsampling), followed by four decoder blocks (transpose convolution and convolution). This U-Net uses spatial pooling between each of its encoder blocks to downsample the input, and transpose convolutions between each of its decoder blocks to upsample intermediate activities. We validated our implementation of this U-Net following the author's training routine, and were able to replicate their reported "superhuman" performance in cell segmentation on SNEMI3D (Fig. S1).

The $\gamma$-Net for connectomics resembles the U-Net architecture of Lee et al. (2017). This $\gamma$-Net replaces the blocks of convolutions and skip connections of that model with a single layer of convolution followed by an fGRU (as in the high level diagram of Fig. 1c, $Conv^{(\ell)}, \rightarrow fGRU^{(\ell)}$). In the encoder pathway, fGRUs store horizontal interactions between spatially neighboring units of the preceding convolutional layer in their hidden states. In the decoder pathway, $\gamma$-Net introduces fGRUs that learn top-down connections between layers, and connect recurrent units from higher-feature processing layers to lower-feature processing ones.

| Name | Tissue | Imaging | Resolution | Voxels (X/Y/Z/Volumes) |
|---|---|---|---|---|
| SNEMI3D | Mouse cortex | mbSEM | $6 \times 6 \times 29$nm | $1024 \times 1024 \times 100 \times 1$ |
| Ding | Mouse retina | SBEM | $13.2 \times 13.2 \times 26$nm | $384 \times 384 \times 384 \times 1$ |

Table S3: SEM image volumes used in membrane prediction. SNEMI3D images and annotations are publicly available (Kasthuri et al., 2015), whereas the Ding dataset is a volume from (Ding et al., 2016) that we annotated.

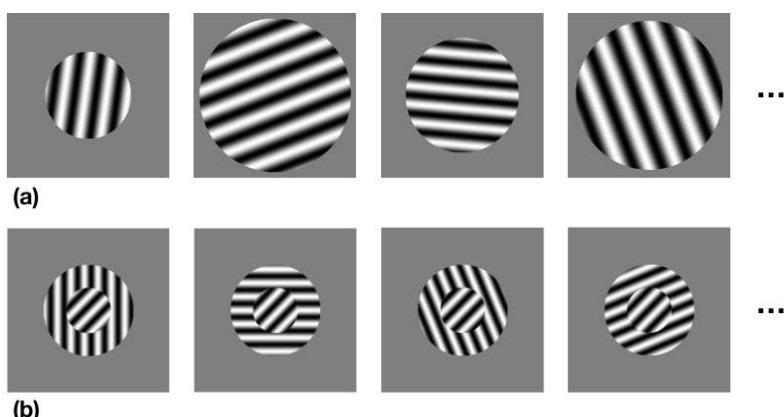

**(a)**

**(b)**

Figure S2: Examples of tilt-illusion stimuli. **(a)** For training images, we sample over a range of size and wavelength to generate single oriented grating patches. **(b)** Test images are obtained by sampling a full range of surround orientation, while fixing all other parameters such as size and frequency of gratings as well as the orientation of the center gratings (at 45 degrees).

Key to the approach of Lee et al. (2017) is their use of a large set of random data augmentations applied to SEM image volumes, which simulate common noise and errors in SEM imaging. These are (i) misalignment between consecutive $z$-locations in each input image volume. (ii) Partial- or fully-missing sections of the input image volumes. (iii) Blurring of portions of the image volume. Augmentations that simulated these types of noise, as well as random flips over the $xyz$-plane, rotations by $90°$, brightness and contrast perturbations, were applied to volumes following the settings of Lee et al. (2017). The model was trained using Adam (Kingma & Ba, 2014) and the learning rate schedule of Lee et al. (2017), in which the optimizer step-size was halved when validation loss stopped decreasing (up to four times). Training involved single-SEM volume batches of $160 \times 160 \times 18$ (X/Y/Z), normalized to $[0, 1]$. As in Lee et al. (2017), models were trained to predict nearest-neighbor voxel affinities, as well as 3 other mid- to long-range voxel distances. Only nearest neighbor affinities were used at test time.

## C  ORIENTATION-TILT ILLUSION IMAGE DATASET

Models were tested for a tilt illusion by first training on grating images of a single orientation, then testing on images in which a center grating had the same/different orientation as a surround grating. Each image in the training dataset consisted of a circular patch of oriented grating on a gray canvas of size $500 \times 500$ pixels. To ensure that the decoder successfully decoded orientation information from model activities, the training dataset incorporated a wide variety of grating stimuli with 4 randomly sampled image parameters: $r$, $\lambda$, $\theta$, and $\phi$. $r$ denotes the radius of the circle in which oriented grating has been rendered, and was sampled from a uniform distribution with interval between 80 and 240 pixels; $\lambda$ specifies the wavelength of the grating pattern and was sampled from a uniform distribution with interval between 30 and 90 pixels; $\theta$ specifies the orientation of the gratings and is uniformly sampled from all possible orientations; $\phi$ denotes the phase offset of the oriented gratings and is also uniformly sampled from all possible values. The models' BSDS-trained weights were fixed and

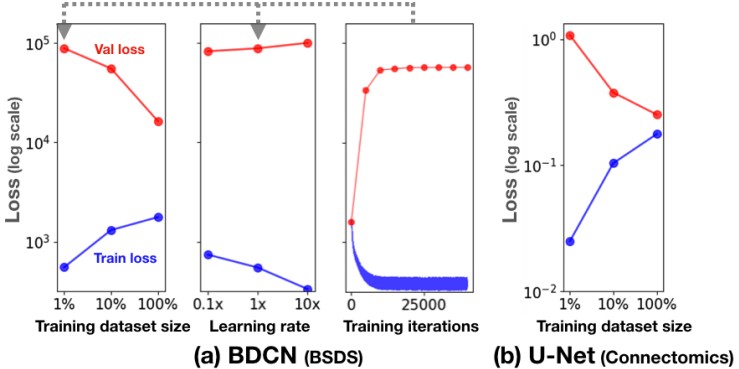

Figure S3: Searching over learning rates did not rescue BSDS performance on small BSDS datasets had no affect on performance. over learning rates did not rescue from overfitting. The left panel depicts training and validation losses for BSDS on different sized subsets of BSDS500. **(b)** Performance after training with three different learning rates on the 5% split. There is little difference in best validation performance between the three learning rates. **(c)** The full training and validation loss curves for the BDCN trained on 5% of BSDS. The model overfits immediately. The model also overfit on the other dataset sizes, but because there was more data, this happened later in training.

readout layers were trained to decode orientation at the center of each image (procedure described in the main text).

This setup allowed us to tease apart the effects of the surround on the representation of orientation in the center by introducing separate surround regions in each test image filled with gratings with same/different orientations as the center (Fig.S2b). Each test image was generated with one additional parameter, $\Delta\theta$ which specified orientation difference of the surround gratings with respect to the center orientation, $\theta$, and was sampled from a uniform distribution with interval between $-90$ and $+90$ degrees. The radius of the surround grating is denoted by $r$ and was sampled from the same uniform distribution we used in training dataset. Center gratings are then rendered in a circle of radius that is one half of the surround gratings.

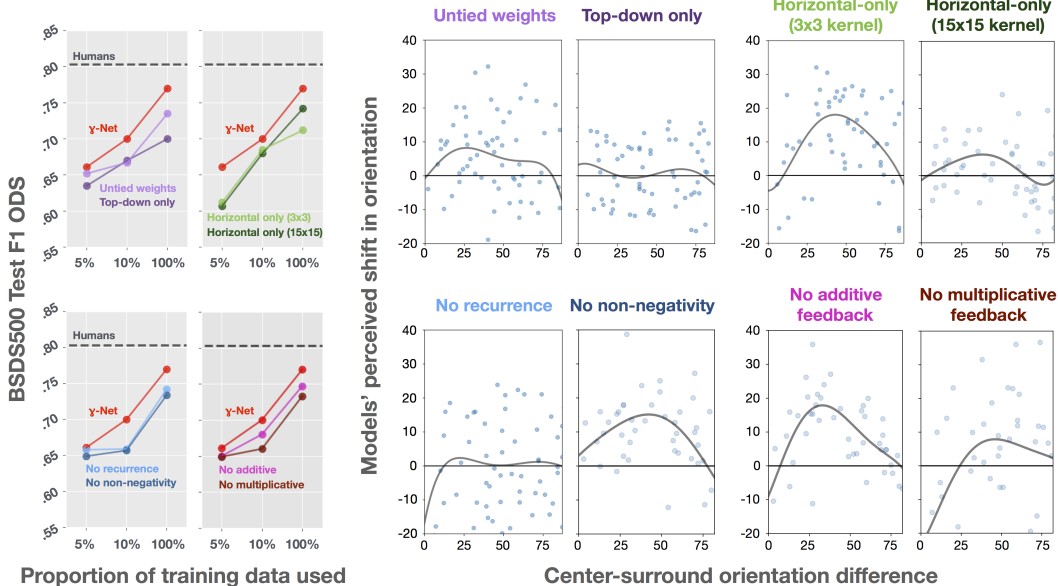

Figure S4: **Performance of lesioned $\gamma$-Nets**. We evaluate the F1 ODS of these models on BSDS500 test images (left) and test for the presence of an orientation-tilt illusion (right). **Top row:** A comparison of $\gamma$-Nets with "untied weights" (i.e., unrolled to 8-timesteps with unique weights on every timestep), only top-down connections, only horizontal connections (using the standard $3\times3$ horizontal kernels), and only horizontal connections but with larger kernels ($15\times15$). The full $\gamma$-Net outperformed each of these models on all subsets of BSDS500 data. Both horizontal-only models captured the repulsive regime of the orientation-tilt illusion when center and surround gratings were similar, but only the version with larger kernels also showed an attractive regime, when the center and surround orientations were dissimilar. **Bottom row:** A comparison of $\gamma$-Nets with no recurrence (i.e., one timestep of processing), no constraint for non-negativity (see fGRU formulation in main text), no parameters for additive feedback ($\mu$ and $\kappa$), and no parameters for multiplicative feedback ($\gamma$ and $\omega$). Once again, the full $\gamma$-Net outperformed each of these models. While none of these models showed the full orientation-tilt illusion, the $\gamma$-Net without non-negativity constraints and the $\gamma$-Net without additive feedback showed the repulsive regime of the orientation-tilt illusion.

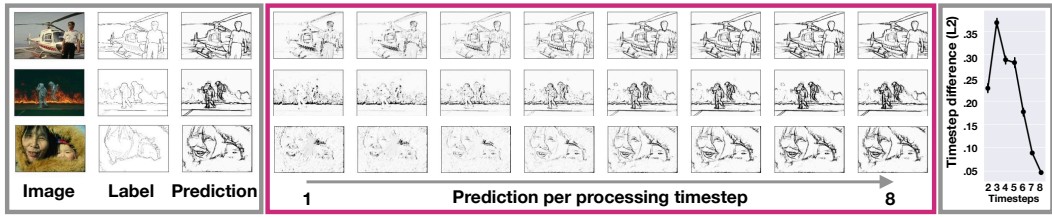

Figure S5: **$\gamma$-Nets trained for contour detection learn to approach a steady-state solution**. The processing timecourse of $\gamma$-Net predictions on representative images from the BSDS500 test set. The L2 norm of per-pixel differences between predictions on consecutive timesteps (i.e., timestep 2 - timestep 1) approaches 0, indicating that the model converges towards steady state by the end of its processing timecourse.

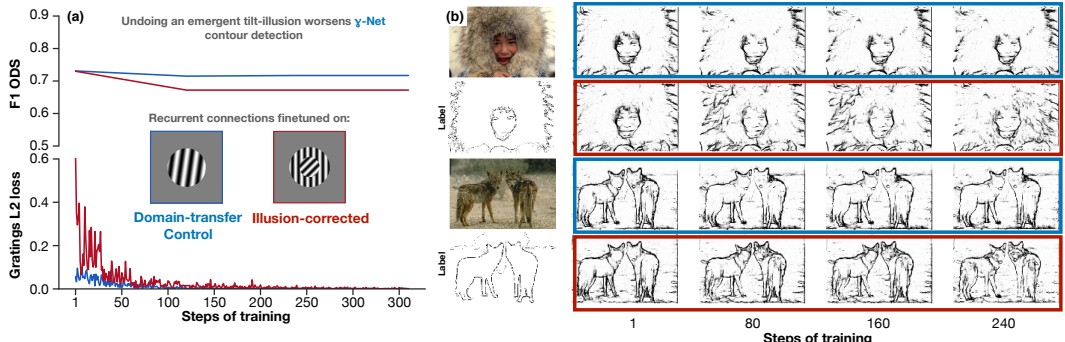

Figure S6: Performance of $\gamma$-Nets during experiments to correct an orientation-tilt illusion. The illusion-corrected model was trained to have veridical representations of the central grating in tilt-illusion stimuli. To control for potential detrimental effects of the training procedure per se, a control model ("domain-transfer control") was trained to decode orientations of single-grating stimuli. (a) Training causes contour-detection performance of both models to drop. However, the illusion-corrected model performance drops significantly more than the biased model (see main text for hypothesis testing). The losses for both models converge towards 0 across training, indicating that both learned to decode central-orientations of their stimuli. (b) Contour detection examples for biased and bias-corrected models across steps of this training procedure.

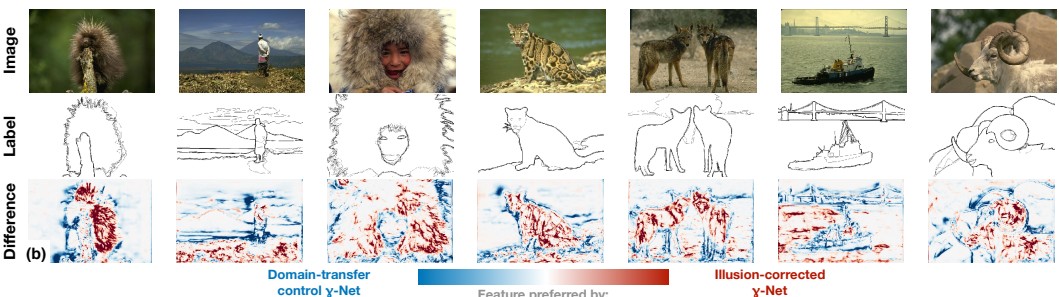

Figure S7: Differences in contour predictions for the illusion-corrected and domain-transfer control $\gamma$-Nets on BSDS500.

