# OpenReview forum: "Recurrent neural circuits for contour detection"
_ICLR.cc/2020/Conference — Accept (Poster)_

### Official Review · AnonReviewer1 · 2019-10-22
**Official Blind Review #1**

**Rating:** 8

**Review:**

The paper introduces a complex hierarchical recurrent model for contour detection loosely inspired by the organization of cortical circuits. Their model performs state-of-the-art on sample-limited versions of popular contour detection (BSDS500) and cell segmentation (SNEMI3D) datasets, and it reproduces the well-known tilt illusion when transfer-learning orientation estimation. Interestingly, "untraining" the tilt illusion degrades performance on contour detection.

Strengths:
+ State-of-the-art performance on data-limited contour detection tasks
+ Nice illustration of how the network refines its predictions over time
+ Demonstrates a contextual visual illusion in a task-trained neural network
+ Shows that tilt illusion is actually necessary for optimal performance in their model

Weaknesses:
- Architecture seems very complicated (unnecessarily so?)
- No ablation studies showing the usefulness of various model components
- Title seems a bit overly general given the quite specific result
- Not clear whether their results support their interpretation of the function of illusions

It's a relatively straightforward paper that is easy to follow and has a clear result that is both interesting and novel. Thus, I'm generally very supportive of the paper.

There are a couple of weaknesses summarised above and detailed below that I would love to see addressed, but none of them is overly critical:

1. Unfortunately the paper suffers from the same issue as the original work on fGRU, which it's based on: The fGRU architecture seems overly complicated and its numerous details and design choices not well motivated. Ablation studies showing which components are really necessary are missing. While this was understandable for the original paper, which introduced a novel approach, one would hope that follow-up work would subsequently get rid of some of the slack and simplify the architecture to the minimum that's really required.

2. The title suggests that the paper explains the function of contextual illusions in general, but the paper actually "just" shows that one contextual illusion emerges when one trains a biologically inspired model on one particular task. I suggest aligning the title better with the actual contribution.

3. (somewhat philosophical) The paper does not really answer the question posed in the abstract, does it? Do visual illusions reflect basic limitations of the visual system or do they correspond to corner cases of neural computations that are efficient in everyday settings? The authors seem argue for the second possibility. But if that was the case, wouldn't one expect other systems trained on the same tasks to also exhibit these illusions? It seems to me as if their results might suggest quite the opposite: Because only brain-like architectures exhibit this illusion, and because only they are hurt by "unlearning" the illusion, this visual illusion may reflect a basic limitation of how the visual system solves the task. I think it would be great if the author could comment on this point and clarify their reasoning in the paper.

**Experience Assessment:**

I have published one or two papers in this area.

**Review Assessment: Checking Correctness Of Derivations And Theory:**

I assessed the sensibility of the derivations and theory.

**Review Assessment: Checking Correctness Of Experiments:**

I carefully checked the experiments.

**Review Assessment: Thoroughness In Paper Reading:**

I read the paper at least twice and used my best judgement in assessing the paper.

---

> ### Author Response · Authors · 2019-11-14
> **Response**
>
> Thank you for your thorough review! We have attempted to address each of your critiques:
>
> <<Architecture seems very complicated>>
> We have included a new model figure, which we believe clarifies the circuitry of the fGRU modules that make up 𝜸-nets (Fig. 1b). We have also included an algorithmic description of how to construct a 𝜸-net (Appendix A), and revised our methods section for clarity. To address your main point, of which parts of the 𝜸-net and fGRU module are necessary or not, we carried additional lesion experiments. In total, we created 𝜸-nets with (a) only top-down connections, (b) only horizontal connections, and (d) untied weights (i.e., with separate weights for every processing timestep, resembling “stacked hourglass” networks). These results can be found in Fig. S4. Only a 𝜸-net with spatially broad horizontal connections showed an orientation-tilt illusion. None of these models matched the performance of the full 𝜸-net on every subset of BSDS500.
>
> We also performed lesion experiments on elements of the fGRUs that make up 𝜸-nets. These include  (a) no non-negativity, (b) no recurrence (i.e., run for 1 timestep), (c) no additive feedback (i.e., \mu and \kappa in the fGRUs are lesioned), and (d) no multiplicative feedback (i.e., \alpha and \omega in the fGRUs are lesioned). These experiments showed that 𝜸-nets perform better when they are recurrent and non-negative. 𝜸-nets also perform better when they have both additive and multiplicative forms of feedback. Notably, the 𝜸-net was better at contour detection and experienced “repulsion” in the orientation-tilt illusion when its additive feedback was lesioned but multiplicative feedback was preserved. When the 𝜸-net’s multiplicative feedback was lesioned but additive feedback was preserved, it showed no hint of the orientation-tilt illusion.
>
> <<Title seems a bit overly general given the quite specific result.>>
> This is a very good point, and we agree that we overreached with the title and some of our discussion. We have downplayed these claims and changed our title.
>
> <<Not clear whether their results support their interpretation of the function of illusions>>
> Our framing for debating contextual illusions as a “bug” or “feature” of biological vision was not very productive. Contextual effects have been extensively studied in visual cortex, and we expect that the circuit mechanisms that give rise to these putatively undesirable effects are nonetheless important for everyday visual perception. We believe that the circuits that drive perception take shortcuts to efficiently process visual features that are relevant to everyday behavior, such as object-boundary contours. Such adaptive biases however have side effects, and this is how we see the orientation-tilt illusion: an undesirable byproduct of efficient strategies for contour detection in the real-world. We see the fact that our 𝜸-nets experienced these illusions as evidence that it is constrained by similar computational biases as neural circuits in visual cortex.

---

### Official Review · AnonReviewer2 · 2019-10-23
**Official Blind Review #2**

**Rating:** 6

**Review:**

The presented paper introduces a novel neural network architecture to explore the question whether visual illusions are corner cases of the human visual system, or whether they represent limitations of perception. The developed recurrent network architecture aims at being more sample efficient than existing methods. The findings discussed in the paper suggest that visual illusions are a byproduct of neural circuits that help to increase the robustness of the human visual system, which in turn suggests that neural networks for processing visual data could benefit from integrating circuits in similar ways. While existing work predominantly aims at explaining whether visual illusions are features or artifacts of the visual system, this work focuses on finding a computational solution to support the hypothesis that visual illusions are features. In particular, the contributions of this work are: (1) novel neural network architecture, called \gamma-networks, which is derived from the work of [Meley et al. 2018] and (2) that the proposed architecture is more sample efficient than SOTA convolutional architectures on contour detection tasks.

Overall, I think this is an interesting paper that can be accepted for publication. However, I am not an expert in this field and am excited to see what the other reviewers think. While the overall contribution of this work appears rather narrow, exploring traits of the human visual system to leverage them in the context of convolutional neural networks for segmentation tasks appears interesting and has the potential to simulate further research in this field. Furthermore, the results are promising and show that the introduced approach is on par with SOTA work in the field. On the downside, many sections of the paper appear unnecessarily cryptic and lack important details. Other sections rely on explanations in the appendix or just refer to related work instead of providing context. This gives me the impression that this work might be better suited for a journal instead of trying to discuss all necessary details into the page limits of ICLR (the paper is also two pages over the common page count). Moreover the topic this paper addresses might be a better fit for a venue that specifically focuses on neuroscience (similar as Mely et al. 2018).


Specific comments:

- Abstract and introduction can be improved by more carefully introducing visual illusions and how the effects of specific traits of the human visual system lead to visual illusions. In its current form abstract and introduction are quite difficult to follow for readers that are less well-versed in neurscience.
- While the main contributions are listed at the end of the introduction, they are not clearly described. It is not clear why it is beneficial that the network generates "an orientation-tilt illusion after it is optimized for contour detection". More details need to be provided here.
- The term "hyper columns" is not properly introduced and needs better motivation, similarly for terms like "suppression" and "facilitation", "circuit integration". Overall it seems the methods section relies too much on the related work for explaining concepts and terminology, which makes this section quite difficult to follow. This needs to be improved.
- It is not clear why it is beneficial to enforce non-negativity. The provided explanation is unclear: "The non-negativity constraints we introduce into the fGRU are necessary to guarantee separate stages of suppression followed by facilitation that can implement asymmetric contextual interactions", which is again because terms were not properly introduced. The term "fGRU" is introduced after it is used.
- Not providing training time with the argument that it is outside the scope of this work seems like a flawed argument and is uncommon practice.
- The datasets listed in Section 4.1. are introduced without references.
- The term "F1 ODS" is not properly introduced.
- Figure 3 (a): the term "Ding" is not introduced. Does this refer to [Ding et al. 2016]?
- The references show many inconsistencies (abbreviated vs. long conference names, etc.). This should be fixed for the final version of this work.
- No limitations are discussed. What are the edge cases in which the method does not perform well?


Typos:
Sec 6: hihg-level

**Experience Assessment:**

I have read many papers in this area.

**Review Assessment: Checking Correctness Of Derivations And Theory:**

I assessed the sensibility of the derivations and theory.

**Review Assessment: Checking Correctness Of Experiments:**

I assessed the sensibility of the experiments.

**Review Assessment: Thoroughness In Paper Reading:**

I read the paper at least twice and used my best judgement in assessing the paper.

---

> ### Author Response · Authors · 2019-11-14
> **Response**
>
> Thank you for your feedback. We hope you will agree that we have addressed your main concerns, and that the paper is far more readable because of it.
>
> <<Abstract and introduction can be improved by more carefully introducing visual illusions and how the effects of specific traits of the human visual system lead to visual illusions. In its current form abstract and introduction are quite difficult to follow for readers that are less well-versed in neuroscience.>>
> We have unpacked neuroscience jargon throughout the paper, and streamlined the abstract and introduction. In particular, we have de-emphasized contextual illusions in the abstract, and extended our discussion of these (and how they emerge) in the main text.
>
> <<While the main contributions are listed at the end of the introduction, they are not clearly described. It is not clear why it is beneficial that the network generates "an orientation-tilt illusion after it is optimized for contour detection". More details need to be provided here.>>
> We have restated our contributions. We hope it is clear that illusions are seemingly undesirable “bugs”. Yet, our model, which shows a similar orientation-tilt illusion as humans, is also far more sample efficient than state-of-the-art models for contour detection that do not experience such an illusion.
>
> <<Overall it seems the methods section relies too much on the related work for explaining concepts and terminology, which makes this section quite difficult to follow. This needs to be improved.>>
> We have rewritten our methods section for clarity, changed our main model figure (see Fig. 1b), and added an algorithmic description of the 𝜸-net architecture (see Appendix A). We hope that these changes will make the methods easier to follow. Please let us know!
>
> <<It is not clear why it is beneficial to enforce non-negativity.>>
> Mely et al. used non-negativity to enforce separate stages for computing recurrent suppression vs. facilitation. This separation is important to ensure that the strength of recurrent suppression— but not facilitation — multiplicatively increases with the net recurrent output. For the orientation-tilt illusion, this means that suppression predominates when center/surround gratings have similar orientations (due to the multiplicative scaling), causing the center grating to be perceived as repulsed from the orientation of the surround. When the center/surround gratings have dissimilar orientations, facilitation predominates because it is additive and not directly scaled by the circuit output. We have clarified this in the methods section.
>
> We have also included a new lesion experiment shown in Fig. S4 to demonstrate that non-negativity is important for the 𝜸-net to experience the orientation-tilt illusion, and improves model sample efficiency in contour detection on BSDS500.
>
> <<The term "fGRU" is introduced after it is used.>>
> We have fixed this.
>
> <<Not providing training time with the argument that it is outside the scope of this work seems like a flawed argument and is uncommon practice.>>
> We apologize for the misunderstanding. We only meant to emphasize that sample efficiency is the goal of this paper. We have added estimates of model wall time during training to Appendix A. As you will see, 𝜸-net models take more time to train than their state-of-the-art counterparts. We have included a new limitations section in our discussion, that mentions this difference in wall time, and we expect that future work on CUDA-level optimizations for 𝜸-net and novel RNN learning algorithms could speed up training time.
>
> <<[The datasets listed in Section 4.1. are introduced without references.][The term "F1 ODS" is not properly introduced.][The term "hyper columns" is not properly introduced and needs better motivation, similarly for terms like "suppression" and "facilitation", "circuit integration".][Figure 3 (a): the term "Ding" is not introduced.][The references show many inconsistencies (abbreviated vs. long conference names, etc.).]>>
> We have addressed each of these issues.

---

### Official Review · AnonReviewer4 · 2019-10-24
**Official Blind Review #4**

**Rating:** 6

**Review:**

The authors propose a new artificial neural network architecture that is derived from a human visual model (Mély et al., 2018). The original (human vision) model can explain some of the human visual illusions, specifically contextual ones. While the adaptation from this human visual model to artificial neural networks was previously done by (Linsley et. al., 2018a), in this paper the authors extend (Linsley et. al., 2018a) to better capture some of the constraints in the human visual model, and also to add a formulation that can also model top-down connections (across layers).
The goal of replicating the structure of the visual human model in artificial neural networks is to improve the machine vision by mimicking the human vision. The results in this paper on contour detection show an improvement regarding sample efficiency with respect to other state of the art methods. Also, the authors show that the artificial model also suffers from visual illusions, and when these are explicitly corrected, its performance drops. This shows that these illusions are a byproduct of the system improving its visual abilities.

Strengths:
1 - Well motivated by cognitive science theory and modeling of the human visual system. The artificial modeling is not just inspired by the human visual system, but derived from an actual human system, replicating it. The paper presents an extension of (Linsley et. al., 2018a), and both the base model and the extension are biologically motivated. The paper also shows that this extension is important to get good results.
2 - Clear structure and ideas. Clearly explained, well written, reasonable ablations and transparent presentation of results, assumptions, limitations, contributions and experiments.
3 - Very good results in contour detection for a very-low data regime, which proves the strength of the model inductive bias. The results show both better performance and good mimicking of the human vision behavior.

Weaknesses
1 – Limited experimental results.
- Is contour detection the only task in which surround suppression helps?
- Lack of experiments in larger contour detection datasets (like Semantic Border Dataset or Cityscapes). Does the model improve accuracy in those, or it just improves sample efficiency in (extremely) small (subsets of) datasets?
- The only results are regarding sample efficiency. Not accuracy, or number of parameters, or running time.
2 – Weak results supporting the claim that the computer vision model mimics the human visual illusions. The authors show an elegant experiment where they explicitly correct the visual illusion and obtain worse results, backing the presented hypothesis. However, the ablation experiments show that the same model without the top-down connections does not present the same results. One would expect that removing a part that is not in the initial formulation from (Mély et al., 2018) should not affect in the visual illusion experiments. Both the explanation at the end of Section 2 and at the sixth paragraph of Section 3 seem to indicate that the _non-negativity_ is the important factor to explain contextual illusions, not the top-down formulation. Can the authors explain whether or not the top-down formulation is a necessary part to model (Mély et al., 2018)? If this is not the case (and top-down is not necessary), the current results would not support the idea that implementing (Mély et al., 2018) in artificial neural networks produces visual illusions in the computer. Also, did the authors perform the visual illusion experiment without the non-negativity? Is it a necessary requisite for the visual illusions to appear? As a positive remark regarding weakness #2, the results still show that the model that performs best is the one having visual illusions.
3 – Related to weaknesses #1 and #2, did the authors perform any experiment on other contextual illusions like the ones explained in (Mély et al., 2018), namely “color induction” or “enhanced color shifts”? Consistent findings across different visual illusions would reinforce the presented hypothesis.

Additional comments:
- While it may not be a "bug", arguing that the visual illusions are a "feature" (both in machine and human visual systems) is probably too much of a claim. At some point the authors refer to it as a "byproduct" of the of the system improving its visual abilities, which I consider a more suitable word.
- For an audience outside of neuroscience, a brief explanation of the concepts “suppression” and “facilitation”, which are very important in the paper, would be convenient.
- Are 8 time-steps sufficient for reaching a steady state? Is the steady state checked in any way?

-------- Updated --------
Rating:
- Weak accept
The authors addressed the initial concerns I raised, providing detailed explanations and additional experiments. While some of these concerns still remain to some extent, I believe this paper explores an interesting direction connecting human visual models with computer vision, obtaining good experimental results. The authors softened some of their initial claims, and the current ones are more supported by the experiments.
Overall, I believe this paper can be a valuable contribution to the conference, and I recommend acceptance.

-------- Previously --------
Rating:
- Weak reject
Overall it is a clear and well-structured paper, with interesting biologically derived architectures (strength #1), but as it stands the experimental results either do not completely support the claims of the paper (weakness #2) or are limited in scope (weakness #1). If the authors can address my concerns, mostly motivating the importance of the experiments for weakness #1, and providing an explanation (possibly correcting me) for weakness #2, I will be happy to increase my rating.

**Experience Assessment:**

I have read many papers in this area.

**Review Assessment: Checking Correctness Of Derivations And Theory:**

I assessed the sensibility of the derivations and theory.

**Review Assessment: Checking Correctness Of Experiments:**

I assessed the sensibility of the experiments.

**Review Assessment: Thoroughness In Paper Reading:**

I read the paper at least twice and used my best judgement in assessing the paper.

---

> ### Author Response · Authors · 2019-11-14
> **Response**
>
> Thank you for the thorough review. To address your comments point-by-point:
>
> <<Is contour detection the only task in which surround suppression helps?>>
> The 𝜸-net and the circuit that it is based on (Mely et al., 2018) are models of long-range contextual interactions in visual cortex, which go beyond surround suppression and contours detection. These models also include mechanisms for surround facilitation and so-called intra-columnar forms of suppression/facilitation (i.e. across features at a retinal location). Mely et al. showed that the circuit which we build our fGRU off of can also explain illusions in color, motion, and disparity domains.
> As we mention in the discussion, we see our work as scaffold linking computer vision with the decades of studies in such classical and extra-classical effects in visual cortex. We expect that future work on our model will also yield similar results for computer vision tasks posed in other domains.
>
> <<Does the model improve accuracy in [datasets like Semantic Border or Cityscapes], or it just improves sample efficiency in (extremely) small (subsets of) datasets?>>
> We have demonstrated that 𝜸-net performs on par (or slightly better) than state-of-the-art approaches (from this year) on contour detection across three different datasets when trained on full, augmented, datasets. We have also showed that 𝜸-nets are far more sample efficient than these approaches, they explain the classical orientation-tilt illusion, and they offer a normative account for why such an illusion might arise. The scope of this work is already pushing the limits of this conference submission guidelines (as pointed out by R2), and we look forward to future work extending our approach to other datasets like the ones suggested here.
>
> <<The only results are regarding sample efficiency. Not accuracy, or number of parameters, or running time.>>
> In the original submission we demonstrated that 𝜸-nets are as accurate (or better) than the state-of-the-art approaches to contour detection when trained with augmentations on three separate datasets. We also mentioned the number of parameters in our 𝜸-net models in Appendix A. We have now also included the number of parameters in the state-of-the-art models that we compare to. The relationship between parameter complexity and sample efficiency in deep hierarchical architectures is hotly debated [1], and our results were somewhat mixed, so we chose not to emphasize it. We have now also added estimates of model wall time during training in Appendix A. As you will see, 𝜸-net models take more time to train than their state-of-the-art counterparts. We have included a new limitations section in our discussion, which mentions these differences in wall time. We believe that this limitation can be resolved by advances in CUDA optimizations and learning algorithms for RNNs.
>
> <<Weak results supporting the claim that the computer vision model mimics the human visual illusions…>>
> Thanks for raising a great point that we did not fully explore in our original submission. The circuit model of Mely et al. explained contextual illusions purely through broad horizontal connections. Indeed, this model used horizontal kernels that were 29x the spatial extent of its feedforward kernels, whereas in the 𝜸-net these kernels have the same spatial extent. As the reviewer mentioned a horizontal-only version of this 𝜸-net exhibited only repulsion in the orientation-tilt illusion, but not the attraction. To test whether this failure to explain the illusion was due to a mismatch between the horizontal kernel sizes of our 𝜸-net vs. the original circuit of Mely et al., we trained a new horizontal-only 𝜸-net with kernels that were 5x the size of its feedforward kernels (consistent with ratios of extra-classical to classical receptive field sizes in primate visual cortex). This broader horizontal-only 𝜸-net showed the full orientation-tilt illusion (at the same time, this model did not perform as well as the original 𝜸-net with both horizontal and top-down connections; see the new Fig. S4).
>
> More generally, top-down feedback is ubiquitous in visual cortex and important for visual recognition tasks. Mely et al. even speculated in their discussion that the spatially wider interactions simulated by their model might be implemented by top-down connections in the brain — a hypothesis that has been verified by recent neurophysiological work [2]. Taken together, it appears that top-down connections are not necessary to exhibit the orientation-tilt illusion, but can do so (a) using fewer parameters, while (b) helping contour detection performance.

---

> > ### Author Response · Authors · 2019-11-14
> > **Response continued**
> >
> > <<Also, did the authors perform the visual illusion experiment without the non-negativity? Is it a necessary requisite for the visual illusions to appear?>>
> > See the new Fig. S4 for contour detection performance and tests for the orientation-tilt illusion in a 𝜸-net without non-negativity. We have also revised our model description to explain why non-negativity is important in our model formulation. Briefly, non-negativity enforces the model to have separate stages for recurrent computing of suppression vs. facilitation, where the strength of recurrent suppression— but not facilitation — multiplicatively increases with the net recurrent output. For the orientation-tilt illusion, this means that suppression predominates when center/surround gratings have similar orientations (due to the multiplicative scaling), causing the center grating to be perceived as repulsed from the orientation of the surround. When the center/surround gratings have dissimilar orientations, facilitation predominates because it is additive and not directly scaled by the circuit output, causing the center grating to be perceived as attracted to the orientation of the surround.
> >
> > <<...did the authors perform any experiment on other contextual illusions like the ones explained in (Mély et al., 2018), namely “color induction” or “enhanced color shifts?>>
> > We leave this question to future work. We suspect that computer vision tasks like color constancy could show similar synergy with these color illusions as we show between contour detection and the orientation-tilt illusion.
> >
> > <<While it may not be a "bug", arguing that the visual illusions are a "feature" (both in machine and human visual systems) is probably too much of a claim.>>
> > We appreciate the feedback. We have revised our title and dialed back our emphasis on this claim.
> >
> > <<For an audience outside of neuroscience, a brief explanation of the concepts “suppression” and “facilitation”, which are very important in the paper, would be convenient.>>
> > We have clarified these concepts and other neuroscience jargon in the manuscript.
> >
> > <<Are 8 time-steps sufficient for reaching a steady state? Is the steady state checked in any way?>>
> > See the new Fig. S5, which depicts 𝜸-net dynamics and approach to a steady-state solution during contour detection on BSDS500.
> >
> > [1] Belkin M, Hsu D, Ma S, & Mandal S. 2019. Reconciling modern machine-learning practice and the classical bias-variance trade-off. PNAS.
> > [2] Chettih S & Harvey C. 2019. Single-neuron perturbations reveal feature-specific competition in V1. Nature.

---

> > > ### Comment · AnonReviewer4 · 2019-11-15
> > > **Thank you, I updated my initial rating**
> > >
> > > Thank you for your detailed rebuttal and additional experiments. I updated my initial rating, please refer to the initial comment.

---

### Author Response · Authors · 2019-11-14
**Response to reviewers**

We thank the reviewers for their detailed feedback. Our revision addresses the main comments of the reviewers, which we believe have greatly improved the work. Please note that we have uploaded two versions of our revised manuscript: (1) The revision, and (2) a diff between this revision and the original submission. Our revised manuscript includes the following changes (requested by the parenthetical reviewers):

- (R1/R4) Figure S4, which shows the results of new experiments to clarify (a) which fGRU mechanisms give rise to an orientation-tilt illusion, and (b) how these mechanisms influence sample efficiency in contour detection. This figure demonstrates the importance of non-negativity, and how a version of the 𝜸-net with only horizontal connections can show an orientation-tilt illusion.
- (R4) Figure S5, depicting 𝜸-net dynamics during contour detection on BSDS-500, which approach steady-state by the end of its processing time-course.
- (R1/R2/R4) We have simplified our model descriptions and explanations, clarified neuroscience terminology, and included a new model figure which we believe is a much clearer depiction of our circuit. We have also included an algorithmic description of the 𝜸-net in Appendix A. Model code is attached to our submission, which we will package into a GitHub repo for the final version of this paper.
- (R1/R2) We have expanded our discussion of the limitations of our work as well as the wall time for our models.
- Fixed typos and inconsistent formatting in our references.

We will also respond to each of the reviewer comments directly. Please let us know if you have any other questions, comments, or concerns that you would like us to address before the response period ends.

---

> ### Author Response · Authors · 2019-11-14
> **Clarification**
>
> You can find the two different versions of the manuscript by clicking on the "Show revisions" link. The diff is the most recent upload, and the revised manuscript is the second most recent upload.

---

### Decision · Program_Chairs · 2019-12-19

**Decision:**

Accept (Poster)

**Comment:**

All the reviewers recommend accept, and the found the paper interesting and novel.